# Large-Scale Adversarial Training for Vision-and-Language Representation Learning

**Zhe Gan[1], Yen-Chun Chen[1], Linjie Li[1], Chen Zhu[2], Yu Cheng[1], Jingjing Liu[1]**
[1]Microsoft Dynamics 365 AI Research,    [2]University of Maryland, College Park
{zhe.gan,yen-chun.chen,lindsey.li,yu.cheng,jingjl}@microsoft.com
chenzhu@cs.umd.edu

## Abstract

We present VILLA, the first known effort on large-scale adversarial training for vision-and-language (V+L) representation learning. VILLA consists of two training stages: ($i$) task-agnostic adversarial pre-training; followed by ($ii$) task-specific adversarial finetuning. Instead of adding adversarial perturbations on image pixels and textual tokens, we propose to perform adversarial training in the embedding space of each modality. To enable large-scale training, we adopt the "free" adversarial training strategy, and combine it with KL-divergence-based regularization to promote higher invariance in the embedding space. We apply VILLA to current best-performing V+L models, and achieve new state of the art on a wide range of tasks, including Visual Question Answering, Visual Commonsense Reasoning, Image-Text Retrieval, Referring Expression Comprehension, Visual Entailment, and NLVR$^2$.[1]

## 1   Introduction

Inspired by the success of BERT [11] on natural language understanding, there has been a surging research interest in developing multimodal pre-training methods for vision-and-language representation learning (*e.g.*, ViLBERT [35], LXMERT [58], and UNITER [10]). When finetuned on downstream tasks, these pre-trained models have achieved state-of-the-art performance across diverse V+L tasks, such as Visual Question Answering (VQA) [4, 15], Visual Commonsense Reasoning (VCR) [72], and Referring Expression Comprehension [69]. However, due to the immense capacity of large-scale pre-trained models yet limited amount of labeled data in downstream tasks, aggressive finetuning often falls into the overfitting trap [22]. *Adversarial training*, a method to combat adversarial attacks in order to create robust neural networks [57, 14], has recently shown great potential in improving the generalization ability of pre-trained language models [76, 22] and image classifiers [64]. A natural question that came to our mind: can we apply similar adversarial training techniques to V+L problems to improve model performance?

We propose VILLA (**Vi**sion-and-**L**anguage **L**arge-scale **A**dversarial training), which advocates the use of adversarial training for V+L representation learning. As illustrated in Figure 1, VILLA consists of two training stages: ($i$) *task-agnostic* adversarial pre-training (APT); followed by ($ii$) *task-specific* adversarial fine-tuning (AFT). Intuitively, if well-designed, multimodal pre-training tasks such as image-conditioned masked language modeling and image-text matching can resonate well with many downstream tasks that require visual grounding and reasoning abilities. This leads to our hypothesis that the improved generalization ability of pre-trained models learned during APT stage can be readily transferred to the AFT stage for diverse tasks. In other words, APT is able to uniformly lift model performance for all downstream tasks in a task-agnostic way, while AFT can further enhance the finetuned models by leveraging task-specific supervision signals.

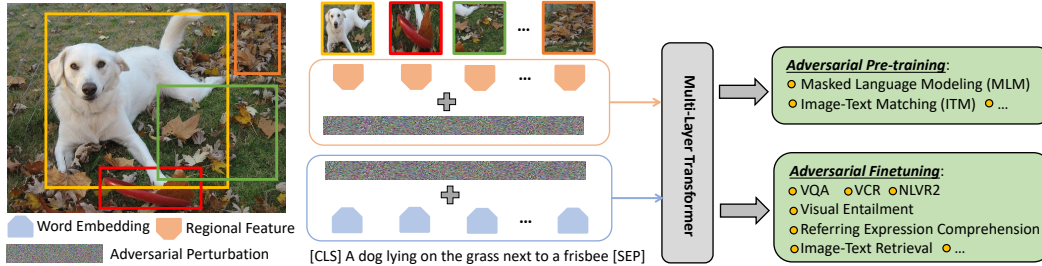

Figure 1: Overview of the proposed VILLA framework for vision-and-language representation learning.

To bring in more flexibility in generating adversarial examples for robust training, we propose to perform adversarial training on the embedding level for multi-modalities, instead of operating on image pixel and sub-word token level in conventional practice. For text modality, we add adversarial perturbations to word embeddings [41, 76, 22]. For image modality, most previous work observes that robustness is at odds with generalization, *i.e.*, trained models are able to resist adversarial attacks on clean images at the expense of performance [39, 65, 74]. Distinctive from these studies, we directly add adversarial perturbations to extracted image-region features [2], as our end goal is the final V+L model performance rather than crafting adversarial image examples. Experiments show that this strategy leads to large performance gain on clean inputs.

Adversarial training procedure is time-consuming and computationally expensive. To power efficient large-scale training, we adopt the recently proposed "free" adversarial training strategy [50, 73, 76], which obtains the gradients of parameters with almost no extra cost when computing the gradients of inputs. In addition to requiring adversarial perturbations to be label-preserving, we also introduce KL-divergence-based regularization to enforce the confidence level of the prediction to be close, characterized by the "dark" knowledge hidden in the probability vectors. This promotes higher smoothness of the training objective and has empirically proven as important regularization effective for further performance boost.

For evaluation, we mostly focus on UNITER [10], the current best-performing V+L model with state-of-the-art performance across many popular V+L benchmarks, and enhance UNITER with VILLA through comprehensive experiments on six V+L tasks: VQA [15], VCR [72], NLVR$^2$ [54], Visual Entailment [66], Referring Expression Comprehension [69], and Image-Text Retrieval [27]. VILLA is a generic framework that can be applied to any multimodal pre-training method. To demonstrate its versatility, we further apply it to LXMERT on VQA, GQA [21], and NLVR$^2$ tasks for generalizability test.

The main contributions are summarized as follows. (*i*) We present VILLA, the first known effort on adversarial pre-training and adversarial finetuning for V+L representation learning. (*ii*) Instead of operating on pixel and word token level, we propose to add adversarial perturbations in the embedding space of multi-modalities, and introduce a smoothness-inducing adversarial regularization term on top of the "free" adversarial training strategy. (*iii*) VILLA achieves new state of the art across six popular V+L tasks. In particular, by relying on standard bottom-up image features only [2], VILLA improves the single-model performance of UNITER-large from 74.02 to 74.87 on VQA, and from 62.8 to 65.7 on VCR. With ensemble, VQA performance is further boosted to 75.85.

## 2 Related Work

**Multimodal Pre-training** ViLBERT [35] and LXMERT [58] are the pioneering works in vision+language pre-training, where two Transformers are used to encode image and text modalities, respectively, then a third Transformer is built on top for multimodal fusion. Compared to this two-stream architecture, recent work such as VL-BERT [53], VisualBERT [31], B2T2 [1], Unicoder-VL [28] and UNITER [10] advocate a single-stream model design, where two modalities are directly fused in early stage. More recent studies leverage multi-task learning [36] to enhance finetuning and use detected image tags [32] to further enhance pre-training. Pixel-BERT [19] proposes to align text with image pixels instead of conventional bottom-up features. Multimodal pre-training has brought leaping advances in vision+language understanding tasks such as VQA and VCR, with great potential in extending to visual captioning [75, 63], visual dialog [43, 62], vision-language naviga-

tion [16, 40], as well as video-and-language representation learning [56, 55, 38, 29]. Recent work [7] also investigates the design of probing tasks to understand the knowledge learned in pre-training.

**V+L Representation Learning** Before multimodal pre-training dominated the scene, there had been a long line of studies on how to learn better V+L representations. Prominent work includes: $(i)$ advanced attention mechanisms [37, 70, 47]; $(ii)$ better multimodal fusion methods [12, 71, 25, 24]; $(iii)$ multi-step reasoning [68, 20, 13]; $(iv)$ incorporation of object relations [49, 44, 6, 30]; and $(v)$ neural module networks for compositional reasoning [3, 23, 18, 9]. In principle, our proposed VILLA framework can be plugged into these "shallower" models. In this paper, we mainly focus on enhancing Transformer-based state-of-the-art models.

**Adversarial Training** Adversarial machine learning is an active research area [57, 14, 5]. Algorithms are developed to either attack existing models by constructing adversarial examples, or train robust models to defend against adversarial attacks. Among existing defense approaches, adversarial training (AT) is a general strategy to empower models with state-of-the-art robustness in different settings [59, 39, 65, 74, 46]. Existing research mostly focuses on AT for image classification, and the general notion is that robustness is often at odds with accuracy. Most recently, [64] shows that model accuracy on clean images can be improved if a separate auxiliary batch norm is used for adversarial examples. There are also some parallel studies on applying AT to language modeling [61] and natural language understanding [41, 76, 22]. Due to growing dominance of large-scale pre-training, very recent work has started to explore adversarial training in the pre-training stage [17, 8, 34]. VILLA is the first known effort that studies AT for V+L tasks and adds adversarial perturbations to both image and word embedding space. We also prove that AT can be effectively incorporated in both pre-training and fine-tuning stages. A more detailed discussion on related work is provided in Appendix.

# 3 Vision-and-Language Large-scale Adversarial Training

There are three key designs that encapsulate VILLA's unique strengths in improving performance and generalization of pre-trained V+L models : $(i)$ Adversarial pre-training *and* fine-tuning; $(ii)$ Adding perturbations in the embedding space; and $(iii)$ Enhanced adversarial training algorithm.

## 3.1 Adversarial Pre-training and Finetuning

We first briefly review the *pretrain-then-finetune* paradigm that has become prevalent in V+L representation learning, then describe our proposed two-stage adversarial training framework.

**Pre-training** Let $\mathcal{D}_p$ denote a pre-training dataset, which consists of image-text pairs $(\boldsymbol{x}_{img}, \boldsymbol{x}_{txt})$. The goal in the pre-training stage is to learn universal image-text representations that are generalizable to different downstream tasks. Take one-stream models [10, 53] as an example. Image and text inputs are first represented as low-dimensional feature vectors $\boldsymbol{z}_{img} = g_{bu}(\boldsymbol{x}_{img})$ and $\boldsymbol{z}_{txt} = g_{emb}(\boldsymbol{x}_{txt})$, where $g_{bu}(\cdot)$ represents a fixed bottom-up image feature extractor [2], and $g_{emb}(\cdot)$ represents a learnable word embedding function. Then, a multi-layer Transformer [60] is applied on top to learn multimodal fusion. The above process can be abbreviated as $\tilde{\boldsymbol{z}}_{img}, \tilde{\boldsymbol{z}}_{txt}, \tilde{\boldsymbol{z}}_{cls} = f_{\boldsymbol{\theta}}(\boldsymbol{x}_{img}, \boldsymbol{x}_{txt})$, where $\tilde{\boldsymbol{z}}_{img}$ and $\tilde{\boldsymbol{z}}_{txt}$ represent the contextualized representations of each image region and each textual token, respectively. Typically, V+L models employ a special [CLS] token whose embedding $\tilde{\boldsymbol{z}}_{cls}$ is considered as the joint V+L representation to be used for downstream tasks. $\boldsymbol{\theta}$ denotes all the learnable parameters including the word embedding matrix.

Let $\boldsymbol{y}$ denote the output supervision signal, which is different across different pre-training tasks. There are three typical pre-training tasks used in most V+L models: $(i)$ Masked Language Modeling (MLM): some tokens in $\boldsymbol{x}_{txt}$ are replaced by special [MASK] tokens, and the goal is to predict the masked tokens $\boldsymbol{y}$ based on surrounding multimodal context; $(ii)$ Masked Region Modeling (MRM): the features of some image regions in $\boldsymbol{x}_{img}$ are replaced by zero vectors, and the goal is to predict the masked image regions $\boldsymbol{y}$ given the remaining multimodal information (via cross-entropy loss, KL-divergence loss [35], or contrastive learning [55]); $(iii)$ Image-Text Matching (ITM): both $\boldsymbol{x}_{img}$ and $\boldsymbol{x}_{txt}$ are kept intact, and the goal is to predict a binary label $\boldsymbol{y}$ to judge whether the input image and text are paired or not.

**Finetuning** Given a downstream task $\mathcal{T}_f$ and a supervised dataset $\mathcal{D}_f$ consisting of $(\boldsymbol{x}_{img}, \boldsymbol{x}_{txt}, \boldsymbol{y})$, the pre-trained model can be finetuned by introducing a small neural network $h(\cdot)$ on top of $\tilde{\boldsymbol{z}}_{cls}$ and minimizing the cross-entropy loss. $\boldsymbol{\theta}$ is initialized with pre-trained weights, and $\boldsymbol{y}$ now becomes

a label. For example, in VQA, $\boldsymbol{y}$ corresponds to the ground-truth answer from a candidate pool, represented as a one-hot vector. In VCR [72], it is a four-way classification label.

In both pre-training and finetuning, by instantiating different $\boldsymbol{y}$, the training process can be uniformly abstracted as an empirical risk minimization problem:

$$\min_{\boldsymbol{\theta}} \mathbb{E}_{(\boldsymbol{x}_{img}, \boldsymbol{x}_{txt}, \boldsymbol{y}) \sim \mathcal{D}} [L(f_{\boldsymbol{\theta}}(\boldsymbol{x}_{img}, \boldsymbol{x}_{txt}), \boldsymbol{y})] . \tag{1}$$

**Two-stage Adversarial Training** Pre-training and finetuning are inherently connected. Independent of the tasks (*e.g.*, MLM, ITM for pre-training, or VQA for finetuning), model training requires the acquisition of essential reasoning skills that can catalyze multimodal fusion for cross-modality joint understanding. For example, in MLM, a masked token 'dog' can be predicted by looking at the image region that contains a dog; and in VQA, when asked whether there is a dog in an image, such visual grounding skills learned through pre-training can be readily applied. We hypothesize that: (*i*) by performing adversarial training in the pre-training stage, the improved generalization ability of a learned model can be beneficial to the finetuning stage; and (*ii*) in the subsequent finetuning stage, where task-specific training signals become available, adversarial finetuning can be applied again to further boost performance. Since pre-training and finetuning share the same mathematical formulation (Eqn. (1)), the same AT algorithm can be adopted in both stages.

## 3.2 Perturbations in the Embedding Space

For the image modality, since state-of-the-art V+L models typically use image features from pre-trained object detectors as input, we add adversarial perturbations in the feature space directly. Note that even though the main difference is simply the noise injecting space, our approach is distinctive from most previous work where perturbations are applied to the pixel space, which is more rigid than fine-grained embedding perturbation. On the other hand, unlike image pixels that are continuous-valued, discrete tokens in the text modality are more difficult to manipulate. It remains unclear how to craft label-preserving adversarial examples without changing the original semantic meaning of the sentence. But since we only care about the *ultimate effects* of adversarial training on downstream tasks, not intepretability of adversarial examples, we choose to add perturbations to the word embeddings following [76].

In pre-trained V+L models, positional embeddings are used to encode the location of image regions and sub-word tokens. Our adversaries only modify image and word embeddings, leaving other components of the multimodal features unchanged. Furthermore, due to the distinct characteristics of image and text modalities, we propose to add perturbations to one modality at a time. Specifically, we add adversarial perturbations $\boldsymbol{\delta}_{img}$ and $\boldsymbol{\delta}_{txt}$ such that the prediction becomes $\hat{\boldsymbol{y}} = f_{\boldsymbol{\theta}}(\boldsymbol{x}_{img} + \boldsymbol{\delta}_{img}, \boldsymbol{x}_{txt})$ and $\tilde{\boldsymbol{y}} = f_{\boldsymbol{\theta}}(\boldsymbol{x}_{img}, \boldsymbol{x}_{txt} + \boldsymbol{\delta}_{txt})$. To preserve original semantics, the norm of $\boldsymbol{\delta}_{img}$ and $\boldsymbol{\delta}_{txt}$ is controlled to be small. Also assumed is that model prediction should not change after the perturbation.

## 3.3 "Free" Multimodal Adversarial Training

**Training Objective** In VILLA, we use adversarial training as an effective regularization to improve model generalization, *i.e.,* to minimize the following objective:

$$\min_{\boldsymbol{\theta}} \mathbb{E}_{(\boldsymbol{x}_{img}, \boldsymbol{x}_{txt}, \boldsymbol{y}) \sim \mathcal{D}} \Big[ \mathcal{L}_{std}(\boldsymbol{\theta}) + \mathcal{R}_{at}(\boldsymbol{\theta}) + \alpha \cdot \mathcal{R}_{kl}(\boldsymbol{\theta}) \Big] , \tag{2}$$

where $\mathcal{L}_{std}(\boldsymbol{\theta}) = L(f_{\boldsymbol{\theta}}(\boldsymbol{x}_{img}, \boldsymbol{x}_{txt}), \boldsymbol{y})$ is the cross-entropy loss on clean data, $\mathcal{R}_{at}(\boldsymbol{\theta})$ is the label-preserving AT loss, and $\mathcal{R}_{kl}(\boldsymbol{\theta})$ is a finer-grained adversarial regularization term. Specifically,

$$\mathcal{R}_{at}(\boldsymbol{\theta}) = \max_{||\boldsymbol{\delta}_{img}|| \leq \epsilon} L(f_{\boldsymbol{\theta}}(\boldsymbol{x}_{img} + \boldsymbol{\delta}_{img}, \boldsymbol{x}_{txt}), \boldsymbol{y}) + \max_{||\boldsymbol{\delta}_{txt}|| \leq \epsilon} L(f_{\boldsymbol{\theta}}(\boldsymbol{x}_{img}, \boldsymbol{x}_{txt} + \boldsymbol{\delta}_{txt}), \boldsymbol{y}) , \tag{3}$$

where $L$ is the cross-entropy loss on adversarial embeddings. Frobenius norm is used to constrain $\boldsymbol{\delta}_{img}$ and $\boldsymbol{\delta}_{txt}$. For optimization, [39] demonstrated that the outer minimization in Eqn. (2) can be solved by SGD, while the inner maximization in Eqn. (3) can be solved reliably by PGD, a standard method for large-scale constrained optimization. Take $\boldsymbol{\delta}_{img}$ for example: PGD takes the following step (with step-size $\alpha$) in each iteration:

$$\boldsymbol{\delta}_{img,t+1} = \Pi_{||\boldsymbol{\delta}_{img}|| \leq \epsilon} (\boldsymbol{\delta}_{img,t} + \alpha g(\boldsymbol{\delta}_{img,t}) / ||g(\boldsymbol{\delta}_{img,t})||_F) , \tag{4}$$

**Algorithm 1** "Free" Multi-modal Adversarial Training used in VILLA.

---

**Require:** Training samples $\mathcal{D} = \{(\boldsymbol{x}_{img}, \boldsymbol{x}_{txt}, \boldsymbol{y})\}$, perturbation bound $\epsilon$, learning rate $\tau$, ascent steps $K$, ascent step size $\alpha$
1: Initialize $\boldsymbol{\theta}$
2: **for** epoch $= 1 \dots N_{ep}$ **do**
3:     **for** minibatch $B \subset X$ **do**
4:         $\boldsymbol{\delta}_0 \leftarrow \frac{1}{\sqrt{N_\delta}} U(-\epsilon, \epsilon), \ \boldsymbol{g}_0 \leftarrow 0$
5:         **for** $t = 1 \dots K$ **do**
6:             Accumulate gradient of parameters $\boldsymbol{\theta}$ given $\boldsymbol{\delta}_{img,t-1}$ and $\boldsymbol{\delta}_{txt,t-1}$
7:             $\boldsymbol{g}_t \leftarrow \boldsymbol{g}_{t-1} + \frac{1}{K} \mathbb{E}_{(\boldsymbol{x}_{img}, \boldsymbol{x}_{txt}, \boldsymbol{y}) \in B}[\nabla_{\boldsymbol{\theta}}(\mathcal{L}_{std}(\boldsymbol{\theta}) + \mathcal{R}_{at}(\boldsymbol{\theta}) + \mathcal{R}_{kl}(\boldsymbol{\theta}))]$
8:             Update the perturbation $\boldsymbol{\delta}_{img}$ and $\boldsymbol{\delta}_{txt}$ via gradient ascend
9:             $\tilde{\boldsymbol{y}} = f_{\boldsymbol{\theta}}(\boldsymbol{x}_{img}, \boldsymbol{x}_{txt})$
10:           $\boldsymbol{g}_{img} \leftarrow \nabla_{\boldsymbol{\delta}_{img}}[L(f_{\boldsymbol{\theta}}(\boldsymbol{x}_{img} + \boldsymbol{\delta}_{img}, \boldsymbol{x}_{txt}), \boldsymbol{y}) + L_{kl}(f_{\boldsymbol{\theta}}(\boldsymbol{x}_{img} + \boldsymbol{\delta}_{img}, \boldsymbol{x}_{txt}), \tilde{\boldsymbol{y}})]$
11:           $\boldsymbol{\delta}_{img,t} \leftarrow \Pi_{\|\boldsymbol{\delta}_{img}\|_F \leq \epsilon}(\boldsymbol{\delta}_{img,t-1} + \alpha \cdot \boldsymbol{g}_{img}/\|\boldsymbol{g}_{img}\|_F)$
12:           $\boldsymbol{g}_{txt} \leftarrow \nabla_{\boldsymbol{\delta}_{txt}}[L(f_{\boldsymbol{\theta}}(\boldsymbol{x}_{img}, \boldsymbol{x}_{txt} + \boldsymbol{\delta}_{txt}), \boldsymbol{y}) + L_{kl}(f_{\boldsymbol{\theta}}(\boldsymbol{x}_{img}, \boldsymbol{x}_{txt} + \boldsymbol{\delta}_{txt}), \tilde{\boldsymbol{y}})]$
13:           $\boldsymbol{\delta}_{txt,t} \leftarrow \Pi_{\|\boldsymbol{\delta}_{txt}\|_F \leq \epsilon}(\boldsymbol{\delta}_{txt,t-1} + \alpha \cdot \boldsymbol{g}_{txt}/\|\boldsymbol{g}_{txt}\|_F)$
14:         **end for**
15:         $\boldsymbol{\theta} \leftarrow \boldsymbol{\theta} - \tau \boldsymbol{g}_K$
16:     **end for**
17: **end for**

---

where $g(\boldsymbol{\delta}_{img,t}) = \nabla_{\boldsymbol{\delta}_{img}} L(f_{\boldsymbol{\theta}}(\boldsymbol{x}_{img} + \boldsymbol{\delta}_{img}, \boldsymbol{x}_{txt}), \boldsymbol{y})$ is the gradient of the loss w.r.t. $\boldsymbol{\delta}_{img}$, and $\Pi_{\|\boldsymbol{\delta}_{img}\| \leq \epsilon}$ performs a projection onto the $\epsilon$-ball.

To further enhance the above AT algorithm, $\mathcal{R}_{kl}(\boldsymbol{\theta})$ is defined as

$$
\mathcal{R}_{kl}(\boldsymbol{\theta}) = \max_{\|\boldsymbol{\delta}_{img}\| \leq \epsilon} L_{kl}(f_{\boldsymbol{\theta}}(\boldsymbol{x}_{img} + \boldsymbol{\delta}_{img}, \boldsymbol{x}_{txt}), f_{\boldsymbol{\theta}}(\boldsymbol{x}_{img}, \boldsymbol{x}_{txt}))
$$
$$
+ \max_{\|\boldsymbol{\delta}_{txt}\| \leq \epsilon} L_{kl}(f_{\boldsymbol{\theta}}(\boldsymbol{x}_{img}, \boldsymbol{x}_{txt} + \boldsymbol{\delta}_{txt}), f_{\boldsymbol{\theta}}(\boldsymbol{x}_{img}, \boldsymbol{x}_{txt})), \quad (5)
$$

where $L_{kl}(p, q) = \text{KL}(p\|q) + \text{KL}(q\|p)$, $p, q$ denote the two probability distributions, and $\text{KL}(\cdot)$ denotes the Kullback-Leibler Divergence. Compared to Eqn. (3) that promotes label-preserving adversarial attack, Eqn. (5) further advocates that the confidence level of the prediction, characterized by the probability vector over the simplex $\Delta_n$ ($n$ is the number of classes), should also be close. Similar techniques are used in Virtual AT [42], TRADES [74], and UDA [67]. However, previous work mostly focuses on semi-supervised learning or trade-off between accuracy and robustness; in our work, we found that it is highly effective for boosting model generalization ability.

**"Free" AT Strategy** $K$-step PGD requires $K$ forward-backward passes through the network, which is computationally heavy. Another limitation is that after $K$ steps, only perturbations at the final step are used for model training. To enable AT for large-scale training and promote diverse adversaries, we follow FreeLB [76] to perform multiple PGD iterations to craft adversarial embeddings, and simultaneously accumulate the "free" parameter gradients $\nabla_{\boldsymbol{\theta}} L$ in each iteration. After that, the model parameters $\boldsymbol{\theta}$ are updated all at once with the accumulated gradients, effectively creating a $K$-times-larger "virtual" mini-batch. The full procedure is provided in Algorithm 1.

## 4 Experiments

### 4.1 Experimental Setting

**Downstream Tasks** To validate the effectiveness of VILLA, we apply it to existing V+L pre-trained models and conduct a comprehensive evaluation over a wide range of downstream tasks, including Visual Question Answering (VQA), Visual Commonsense Reasoning (VCR), Referring Expression (RE) Compression, Visual Entailment, Image-Text Retrieval, and NLVR$^2$. To validate the strength of VILLA in model pre-training and finetuning, we first incorporate it into state-of-the-art UNITER model in both stages for downstream evaluation and ablation analysis. And to demonstrate the versatility of VILLA, we further apply it to another V+L model LXMERT [58] with a different architecture design from UNITER (two-stream vs. one-stream) for generalizability test.

| Method | VQA | | VCR | | | NLVR$^2$ | | SNLI-VE | |
|---|---|---|---|---|---|---|---|---|---|
| | test-dev | test-std | Q→A | QA→R | Q→AR | dev | test-P | val | test |
| ViLBERT | 70.55 | 70.92 | 72.42 (73.3) | 74.47 (74.6) | 54.04 (54.8) | - | - | - | - |
| VisualBERT | 70.80 | 71.00 | 70.8 (71.6) | 73.2 (73.2) | 52.2 (52.4) | 67.4 | 67.0 | - | - |
| LXMERT | 72.42 | 72.54 | - | - | - | 74.90 | 74.50 | - | - |
| Unicoder-VL | - | - | 72.6 (73.4) | 74.5 (74.4) | 54.4 (54.9) | - | - | - | - |
| 12-in-1 | 73.15 | - | - | - | - | - | 78.87 | - | 76.95 |
| VL-BERT$_{BASE}$ | 71.16 | - | 73.8 (-) | 74.4 (-) | 55.2 (-) | - | - | - | - |
| Oscar$_{BASE}$ | 73.16 | 73.44 | - | - | - | 78.07 | 78.36 | - | - |
| UNITER$_{BASE}$ | 72.70 | 72.91 | 74.56 (75.0) | 77.03 (77.2) | 57.76 (58.2) | 77.18 | 77.85 | 78.59 | 78.28 |
| VILLA$_{BASE}$ | **73.59** | **73.67** | **75.54 (76.4)** | **78.78 (79.1)** | **59.75 (60.6)** | **78.39** | **79.30** | **79.47** | **79.03** |
| VL-BERT$_{LARGE}$ | 71.79 | 72.22 | 75.5 (75.8) | 77.9 (78.4) | 58.9 (59.7) | - | - | - | - |
| Oscar$_{LARGE}$ | 73.61 | 73.82 | - | - | - | 79.12 | 80.37 | - | - |
| UNITER$_{LARGE}$ | 73.82 | 74.02 | 77.22 (77.3) | 80.49 (80.8) | 62.59 (62.8) | 79.12 | 79.98 | 79.39 | 79.38 |
| VILLA$_{LARGE}$ | **74.69** | **74.87** | **78.45 (78.9)** | **82.57 (82.8)** | **65.18 (65.7)** | **79.76** | **81.47** | **80.18** | **80.02** |

(a) Results on VQA, VCR, NLVR$^2$, and SNLI-VE.

| Method | RefCOCO+ | | | | | | RefCOCO | | | | | |
|---|---|---|---|---|---|---|---|---|---|---|---|---|
| | val | testA | testB | val$^d$ | testA$^d$ | testB$^d$ | val | testA | testB | val$^d$ | testA$^d$ | testB$^d$ |
| ViLBERT | - | - | - | 72.34 | 78.52 | 62.61 | - | - | - | - | - | - |
| VL-BERT$_{BASE}$ | 79.88 | 82.40 | 75.01 | 71.60 | 77.72 | 60.99 | - | - | - | - | - | - |
| UNITER$_{BASE}$ | 83.66 | 86.19 | 78.89 | 75.31 | 81.30 | 65.58 | 91.64 | 92.26 | 90.46 | 81.24 | 86.48 | 73.94 |
| VILLA$_{BASE}$ | **84.26** | **86.95** | **79.22** | **76.05** | **81.65** | **65.70** | **91.93** | **92.79** | **91.38** | **81.65** | **87.40** | **74.48** |
| VL-BERT$_{LARGE}$ | 80.31 | 83.62 | 75.45 | 72.59 | 78.57 | 62.30 | - | - | - | - | - | - |
| UNITER$_{LARGE}$ | 84.25 | **86.34** | 79.75 | 75.90 | 81.45 | 66.70 | 91.84 | 92.65 | 91.19 | 81.41 | 87.04 | 74.17 |
| VILLA$_{LARGE}$ | **84.40** | 86.22 | **80.00** | **76.17** | **81.54** | **66.84** | **92.58** | **92.96** | **91.62** | **82.39** | **87.48** | **74.84** |

(b) Results on RefCOCO+ and RefCOCO. The superscript $d$ denotes evaluation using detected proposals.

| Method | RefCOCOg | | | | Flickr30k IR | | | Flickr30k TR | | |
|---|---|---|---|---|---|---|---|---|---|---|
| | val | test | val$^d$ | test$^d$ | R@1 | R@5 | R@10 | R@1 | R@5 | R@10 |
| ViLBERT | - | - | - | - | 58.20 | 84.90 | 91.52 | - | - | - |
| Unicoder-VL | - | - | - | - | 71.50 | 90.90 | 94.90 | 86.20 | 96.30 | 99.00 |
| UNITER$_{BASE}$ | 86.52 | 86.52 | 74.31 | 74.51 | 72.52 | 92.36 | **96.08** | 85.90 | 97.10 | 98.80 |
| VILLA$_{BASE}$ | **88.13** | **88.03** | **75.90** | **75.93** | **74.74** | **92.86** | 95.82 | **86.60** | **97.90** | **99.20** |
| UNITER$_{LARGE}$ | 87.85 | 87.73 | 74.86 | 75.77 | 75.56 | 94.08 | 96.76 | 87.30 | **98.00** | **99.20** |
| VILLA$_{LARGE}$ | **88.42** | **88.97** | **76.18** | **76.71** | **76.26** | **94.24** | **96.84** | **87.90** | 97.50 | 98.80 |

(c) Results on RefCOCOg and Flickr30k Image Retrieval (IR) and Text Retrieval (TR).

Table 1: Comparison with state-of-the-art pre-trained models on all the downstream tasks.

**UNITER and LXMERT** UNITER-base is a single-stream model, which has 12 layers, with 768 hidden units per layer and 12 attention heads; UNITER-large has 24 layers, with 1024 hidden units per layer and 16 attention heads. UNITER shares the same structure as BERT, except that the input now becomes a mixed sequence of two modalities. LXMERT is a two-stream model, which first performs self-attention through several layers on each modality independently (9 layers for text modality, and 5 layers for image modality), then fuses the outputs of both streams through another 5 layers (first cross-attention, then self-attention).

**Implementation Details** For UNITER experiments, we pre-train with the same four large-scale datasets used in the original model: COCO [33], Visual Genome (VG) [26], Conceptual Captions [52] and SBU Captions [45]. VILLA is applied to both MLM and ITM pre-training tasks. The original UNITER-base (12 layers) and UNITER-large (24 layers) models take 200k and 500k steps for pre-training, respectively. For fair comparison, when applying VILLA to UNITER-base, we run 100k steps of standard training, followed by 100k steps of adversarial training. When applying VILLA to UNITER-large, to save pre-training time,[2] we run 425k steps of standard training, followed by 75k steps of adversarial training.

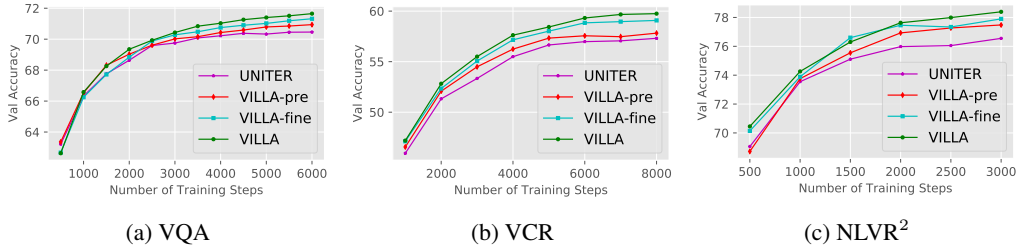

| | (a) VQA | (b) VCR | (c) NLVR$^2$ |

Figure 2: The training curves of VILLA and UNITER on different tasks. For VQA, an internal val set is used.

| Method | VQA | VCR (val) | | | NLVR$^2$ | VE | Flickr30k IR | | | RefCOCO | | Ave. |
|---|---|---|---|---|---|---|---|---|---|---|---|---|
| | test-dev | Q→A | QA→R | Q→AR | test-P | test | R@1 | R@5 | R@10 | testA$^d$ | testB$^d$ | |
| UNITER (reimp.) | 72.70 | 74.24 | 76.93 | 57.31 | 77.85 | 78.28 | 72.52 | 92.36 | 96.08 | 86.48 | 73.94 | 78.06 |
| VILLA-pre | 73.03 | 74.76 | 77.04 | 57.82 | 78.44 | 78.43 | 73.76 | 93.02 | 96.28 | 87.34 | 74.35 | 78.57 |
| VILLA-fine | 73.29 | 75.18 | 78.29 | 59.08 | 78.84 | 78.86 | 73.46 | 92.98 | 96.26 | 87.17 | 74.31 | 78.88 |
| VILLA | 73.59 | 75.54 | 78.78 | 59.75 | 79.30 | 79.03 | 74.74 | 92.86 | 95.82 | 87.40 | 74.48 | **79.21** |

Table 2: Ablation study on VILLA-pre (pre-training) and VILLA-fine (finetuning) with base model size.

| Method | VQA | VCR (val) | | |
|---|---|---|---|---|
| | test-dev | Q→A | QA→R | Q→AR |
| VILLA$_{BASE}$ (txt) | 73.50 | 75.60 | 78.70 | 59.67 |
| VILLA$_{BASE}$ (img) | 73.50 | **75.81** | 78.43 | 59.68 |
| VILLA$_{BASE}$ (both) | **73.59** | 75.54 | **78.78** | **59.75** |
| VILLA$_{LARGE}$ (txt) | 74.55 | 78.08 | 82.31 | 64.63 |
| VILLA$_{LARGE}$ (img) | 74.46 | 78.08 | 82.28 | 64.51 |
| VILLA$_{LARGE}$ (both) | **74.69** | **78.45** | **82.57** | **65.18** |

| Method | VQA | VCR (val) | | |
|---|---|---|---|---|
| | test-dev | Q→A | QA→R | Q→AR |
| UNITER$_{BASE}$ (reimp.) | 72.70 | 74.24 | 76.93 | 57.31 |
| UNITER$_{BASE}$+FreeLB | 72.82 | 75.13 | 77.90 | 58.73 |
| VILLA$_{BASE}$-fine | **73.29** | **75.49** | **78.34** | **59.30** |
| UNITER$_{LARGE}$ (reimp.) | 73.82 | 76.70 | 80.61 | 62.15 |
| UNITER$_{LARGE}$+FreeLB | 73.87 | 77.19 | 81.44 | 63.24 |
| VILLA$_{LARGE}$-fine | **74.32** | **77.75** | **82.10** | 63.99 |

| (a) Image vs. Text Modality. | (b) FreeLB vs. VILLA. |

Table 3: Ablation study on adding perturbations to different modalities and on the VILLA algorithm.

## 4.2 Results and Ablation Analysis

**Downstream Task Evaluation** Table 1 summarizes the results of VILLA applied to UNITER on all evaluation tasks. Compared with existing pre-trained V+L models, our VILLA method achieves new state of the art across all the benchmarks. Specifically, VILLA-base model outperforms UNITER-base by +0.76 on *VQA*, +2.4 on *VCR* for Q→AR, +1.45 on *NLVR$^2$*, +0.75 on *SNLI-VE*, +2.22/+0.70 on *Flickr30k* for Image/Text Retrieval (R@1), and +0.99 on average for the three *RE* datasets.

Similar universal performance lift is also observed in VILLA-large. It is highly encouraging to see that VILLA-large brings an absolute +2.9 points performance gain over UNITER-large for VCR on the Q→AR metric. Compared to the others, VCR is a relatively more challenging task, which requires commonsense reasoning and understanding complex social dynamics that is implicitly encoded in the image. Another significant boost is over the well-studied VQA benchmark, from 74.02 to 74.87. With ensemble, the performance of VILLA-large is further lifted to 75.85.

**Pre-training vs. Finetuning** To understand the effects of adversarial training on pre-training and finetuning, we conduct an ablation study with UNITER-base and summarize the results in Table 2. UNITER (reimp.) denotes our re-implementation of the UNITER-base model with standard training. VILLA-pre and VILLA-fine apply adversarial training to only the pre-training or finetuning stage, respectively. Averaged over the six evaluation tasks, VILLA-pre and VILLA-fine brings +0.51 and +0.82 points performance gain. By combining the two, +1.15 points gain is achieved. Figure 2 further provides the training curves of each task, which illustrate growing performance gaps between AT-enhanced models and the original UNITER, as the number of training steps increases. Interestingly, on VQA, though in early epochs UNITER achieves better performance than VILLA, VILLA catches up quickly after a few hundred of steps, which demonstrates the beneficial regularization effect of adversarial training. More training curves on other tasks can be found in Appendix.

To further understand the importance of adversarial pre-training, we use VQA as the probing task, and compare the performance of standard and adversarial pre-training at each intermediate model

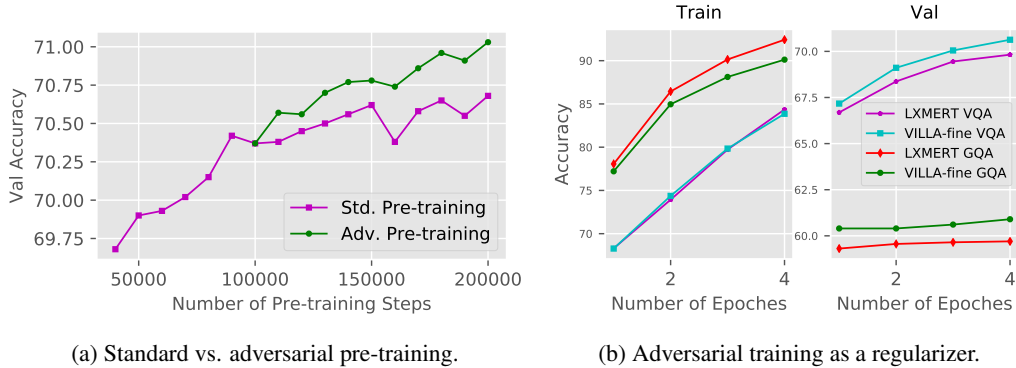

(a) Standard vs. adversarial pre-training.　　　　(b) Adversarial training as a regularizer.

Figure 3: For (a), we use VQA as probing, and compare the performance of standard and adversarial pre-training. For (b), we plot the training curves of standard and adversarial finetuning using LXMERT as backbone.

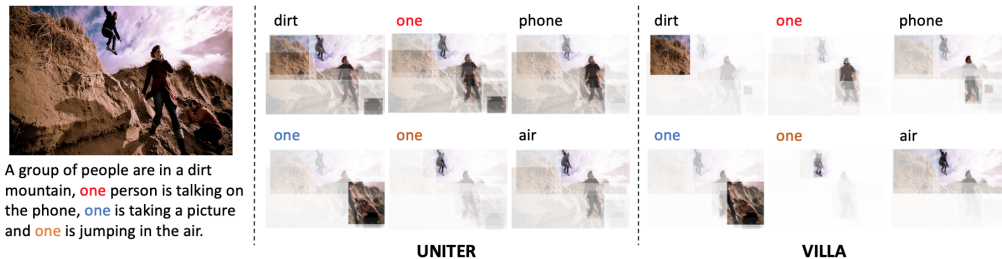

A group of people are in a dirt mountain, one person is talking on the phone, one is taking a picture and one is jumping in the air.

Figure 4: Visualization of text-to-image attention, comparing VILLA against UNITER.

| Model | Visual Coreference (Flickr30k) | | | | | Visual Relation (Visual Genome) | | | | | Ave. |
|---|---|---|---|---|---|---|---|---|---|---|---|
| | scene | clothing | animals | instruments | vehicles | on | standing in | wearing | holding | covering | |
| UNITER$_{BASE}$ | 0.151 | 0.157 | 0.285 | 0.244 | 0.194 | 0.154 | 0.107 | 0.311 | 0.200 | 0.151 | 0.195 |
| VILLA$_{BASE}$ | **0.169** | **0.185** | **0.299** | **0.263** | **0.202** | **0.201** | **0.120** | **0.353** | **0.241** | **0.192** | **0.223** |

Table 4: Probing analysis of the attention heads in pre-trained UNITER and VILLA models.

checkpoint (using standard finetuning to both pre-trained models). Results are presented in Figure 3a. As shown, once adversarial training is activated, VILLA-pre starts outperforming UNITER, and the performance gap increases as the number of pre-training steps grows.

**Image vs. Text Modality** To gain insights on the effects of adversarial examples in different modalities, we conduct experiments by adding perturbations to either image or text modality, and use VQA and VCR for ablation tests. Results are summarized in Table 3a. Conventionally, adversarial training in the image domain hurts model accuracy on clean images. However, in our setting, we observe that adding perturbations to image features alone can boost final model performance significantly. Our initial intuition was that adding perturbations to both modalities might increase the diversity of adversarial examples, hence bringing more benefits. However, ablation results show that adding perturbations on one modality is already gaining significant improvement.[3] The boost on VCR is larger than VQA, which we hypothesize is due to the higher complexity in VCR task, which adding more adversaries to model training can effectively help.

**FreeLB vs. VILLA** To compare with prior work FreeLB, we conduct an additional ablation study also on VQA and VCR, two representative and challenging V+L tasks. Table 3b shows that VILLA achieves consistently better performance than FreeLB over both benchmarks, thanks to the additional fine-grained adversarial regularization term. For example, FreeLB brings little performance boost on VQA, while VILLA achieves considerable improvement over the baseline.

**Probing Analysis** Pre-trained models are expected to learn intricate knowledge about multimodality correlations, such as visual coreference (*i.e.*, region-phrase alignment) and visual relation (*i.e.*, region-

| Method | VQA | | GQA | | NLVR$^2$ | | Meta-Ave. |
|---|---|---|---|---|---|---|---|
| | test-dev | test-std | test-dev | test-std | dev | test-P | |
| LXMERT | 72.42 | 72.54 | 60.00 | 60.33 | 74.95 | 74.45 | 69.12 |
| LXMERT (reimp.) | 72.50 | 72.52 | 59.92 | 60.28 | 74.72 | 74.75 | 69.12 |
| VILLA-fine | **73.02** | **73.18** | **60.98** | **61.12** | **75.98** | **75.73** | **70.00** |

Table 5: Results on LXMERT with VILLA-fine (finetuning).

| Data split | MUTAN | BUTD | BUTD+CC | Pythia | Pythia+CC | BAN | BAN+CC | UNITER | VILLA |
|---|---|---|---|---|---|---|---|---|---|
| Original | 59.08 | 61.51 | 62.44 | 64.08 | 64.52 | 64.97 | 65.87 | 70.35 | **71.27** |
| Rephrasing | 46.87 | 51.22 | 52.58 | 54.20 | 55.65 | 55.87 | 56.59 | 64.56 | **65.35** |

Table 6: Results on VQA-Rephrasings. Both UNITER and VILLA use the base model size. Baseline results are copied from [51].

region interaction). To provide a more direct measurement on how well our adversarial pre-trained model captures such multimodal signals, we conduct a probing analysis following [7]. We consider five most common visual coreference types in Flickr30k Entities [48] and top five visual relations in Visual Genome [26] (listed in Table 4), and calculate the attention weights between region and phrase (or between regions) learned by pre-trained models. Results show that VILLA presents higher attention weights across all the ten categories (0.223 vs. 0.195 on average), indicating a higher probability of identifying those relations. Figure 4 further provides a visualization of text-to-image attention, where VILLA exhibits more accurate and sharper multimodal alignment.

**Results on LXMERT** VILLA is a generic framework that can be readily applied to any V+L models. To demonstrate its generalization ability, we conduct additional experiments using LXMERT as the backbone. Since adversarial pre-training is highly time-consuming, we only focus on adversarial finetuning for LXMERT.[4] We use VQA, GQA and NLVR$^2$ as the evaluation tasks, the same as LXMERT. Results in Table 5 show that VILLA-fine instantly provides +0.88 average performance boost across the three tasks. The training curves are provided in Figure 3b. Compared to LXMERT, VILLA-fine achieves higher accuracy on validation set and lower accuracy on training set for both VQA and GQA, clearly demonstrating its regularization effect in preventing overfitting of large-scale pre-trained models.

**Robustness** In order to test adversarial robustness, we need to perform adversarial attacks to existing V+L models. This V+L attack problem is largely unexplored in the literature. For example, how to reliably back-propagate the gradients from the multimodal Transformer to the CNN backbone to generate image adversaries is non-trivial. How to craft textual adversaries that align with the visual context is also challenging. In this work, we mainly focus on improving model's generalization performance on clean data, leaving a more thorough investigation of adversarial attack and robustness as important future work.

As a proxy for robustness evaluation, we conduct additional experiments on the VQA-Rephrasings dataset [51] to test the robustness of existing V+L models to paraphrases. For fair comparison, we have re-trained both UNITER and VILLA on the VQA training set only. Results are summarized in Table 6, where 'Original' and 'Rephrasing' denote the test set with original questions and their rephrasings, respectively. UNITER has already lifted up the performance by a large margin, and VILLA facilitates further performance boost.

We provide additional experimental results, more details about the probing analysis, and additional visualization examples in Appendix.

## 5   Conclusion

In this paper, we present VILLA, an advanced adversarial training (AT) framework for better vision-and-language representation learning. By performing AT in both pre-training and finetuning stages, and by adding adversarial perturbations to the embedding space, VILLA achieves consistent performance boost on all the benchmarks evaluated. As AT is time-consuming, for future work, we plan to study how to accelerate AT so that it can be more feasible for large-scale pre-training in practice.

## Broader Impact

Our research advances vision-and-language representation learning by incorporating adversarial training in both pre-training and finetuning stages. By utilizing the enormous amount of image-text data available on the web for pre-training, VILLA can absorb multimodal clues to capture multi-channel signals from the world, towards a smarter AI system. Furthermore, VILLA can provide instant performance boost in finetuning stage, which will help accelerate future studies in this field. However, in order to train models to learn such capabilities, our method also calls for a high demand on computational resources due to large-scale training, which could be costly both financially and environmentally. As part of our research effort, we will release our pre-trained models to facilitate future research, to empower others' scientific exploration and save environmental cost.

## Acknowledgments and Disclosure of Funding

We appreciate anonymous reviewers for their constructive feedback. This work was fully supported by Microsoft, and was not funded by any other agencies.

## Footnotes

[1]Code is available at `https://github.com/zhegan27/VILLA`.

[2]VILLA is $K$ times computationally heavier than UNITER, where $K$ is the number of adversarial training steps. We typically select adversarial learning rate from {1e-2, 1e-3}, adversarial training steps to 3, and $\alpha$ (Eqn. 2) from 1.0, 1.5, 2.0. More implementation details are provided in Appendix.

[3]We also tried adding adversarial perturbations to both modalities simultaneously instead of alternatively. Empirically, we observe that they obtained similar performance.

[4]Code is available at `https://github.com/zhegan27/LXMERT-AdvTrain`.

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
