[Supplementary Material]

# Large-Scale Adversarial Training for Vision-and-Language Representation Learning: Supplementary Material

**Zhe Gan**[1], **Yen-Chun Chen**[1], **Linjie Li**[1], **Chen Zhu**[2], **Yu Cheng**[1], **Jingjing Liu**[1]
[1]Microsoft Dynamics 365 AI Research,    [2]University of Maryland, College Park
{zhe.gan,yen-chun.chen,lindsey.li,yu.cheng,jingjl}@microsoft.com
chenzhu@cs.umd.edu

## A  Appendix

This supplementary material contains three sections. Section A.1 reviews additional related work. Section A.2 provides additional experimental results. Section A.3 describes downstream tasks and implementation details.

### A.1  Additional Related Work

**Adversarial Training**  Many efforts have been devoted to improving AT from different angles: ($i$) use triplet-wise metric learning [8, 7] and optimal transport [20] to leverage inter-sample interactions; ($ii$) exploit extra unlabeled training data [12, 1]; and ($iii$) accelerate the training procedure [11, 19, 14]. Specifically, adversarial examples have been explored primarily in the image domain, and only recently started to gain attention in vision-and-language research. [2, 16] studied how to craft adversarial examples for image captioning, and [10] investigated how to derive adversarial rules to attack VQA systems. Different from these studies, we are not interested in crafting actual adversarial examples, but aim to apply AT to improve the final model performance over V+L tasks. Note that "adversarial regularization" was proposed in [9]; however, it is mainly used to overcome the language priors in VQA, which is entirely different from the AT used here.

### A.2  Additional Results

**Results on VQA**  In Table 1a of the main text, we have reported the experimental results on the test-dev and test-std splits of VQA. More detailed results on each question type are provided in Table 1. As shown, VILLA improves over UNITER on all the question types.

| Method | test-dev | | | | test-std | | | |
|---|---|---|---|---|---|---|---|---|
| | yes/no | number | other | overall | yes/no | number | other | overall |
| UNITER_BASE (reimp.) | 88.97 | 55.67 | 62.81 | 72.77 | - | - | - | - |
| VILLA_BASE | 89.37 | 56.86 | 63.90 | 73.59 | 89.41 | 56.78 | 63.84 | 73.67 |
| UNITER_LARGE (reimp.) | 90.13 | 57.24 | 63.70 | 73.86 | - | - | - | - |
| VILLA_LARGE | 90.76 | 58.26 | 64.67 | 74.69 | 90.85 | 57.3 | 64.98 | 74.87 |
| VILLA_LARGE (Ensemble) | 91.24 | 59.73 | 65.98 | 75.68 | 91.30 | 59.23 | 66.20 | 75.85 |

Table 1: More detailed results on VQA.

**Training Curves**  In Figure 3a of the main text, we have provided the training curves on three datasets. The training curves for the remaining three datasets are shown in Figure 1 with similar trend observed.

|            | (a) SNLI-VE | (b) Flickr30k IR | (c) RefCOCO |

Figure 1: Additional training curves of VILLA and UNITER on different tasks.

| Method | VQA test-dev | VCR (val) Q→A | QA→R | Q→AR | Ave. |
|---|---|---|---|---|---|
| UNITER (reimp.) | 73.82 | 76.70 | 80.61 | 62.15 | 72.32 |
| VILLA-pre | 74.05 | 77.16 | 81.02 | 62.99 | 73.80 |
| VILLA-fine | 74.48 | 77.74 | 81.91 | 64.00 | 74.53 |
| VILLA | 74.69 | 78.45 | 82.57 | 65.18 | **75.22** |

Table 2: Ablation study on VILLA-pre (pre-training) and VILLA-fine (finetuning) with large model size.

| Method | VQA (test-dev) 100k | 200k (from scratch) |
|---|---|---|
| UNITER (reimp.) | 72.70 | - |
| VILLA-pre | 73.03 | 73.18 |
| VILLA | 73.59 | 73.69 |

Table 3: Adversarial pre-training from scratch with base model size.

| Method | test-dev Accuracy | Binary | Open | Validity | Plausibility | Consistency | Distribution |
|---|---|---|---|---|---|---|---|
| LXMERT (reimp.) | 59.92 | 77.32 | 44.61 | 97.10 | 85.26 | 89.55 | 1.15 |
| VILLA-fine | 60.98 | 78.17 | 45.86 | 97.07 | 85.44 | 91.09 | 1.20 |

| Method | test-std Accuracy | Binary | Open | Validity | Plausibility | Consistency | Distribution |
|---|---|---|---|---|---|---|---|
| LXMERT (reimp.) | 60.28 | 77.14 | 45.40 | 96.33 | 84.46 | 89.45 | 5.38 |
| VILLA-fine | 61.12 | 78.07 | 46.16 | 96.36 | 84.80 | 91.13 | 5.55 |

Table 4: More detailed results on GQA.

**Pre-training vs. Finetuning with Large Model Size** In Table 2 of the main text, we provided ablation study on adversarial pre-training and finetuning with UNITER-base model size (12 layers). In Table 2, we provide additional ablation study with large model size (24 layers) on a selective set of tasks (VQA and VCR). On average, adversarial pre-training and finetuning bring +1.48 and +2.21 performance gain, respectively. Combining the two AT stages provides further improvement.

**Results on GQA** In Table 5 of the main text, we have reported LXMERT results on GQA enhanced by VILLA-fine. The complete results are provided in Table 4 for reference.

**Adversarial pre-training from scratch** Instead of performing adversarial pre-training from 100k steps, we also conducted experiments on adversarial pre-training from scratch with base model size. Preliminary results on VQA are shown in Table 3. Adversarial pre-training from scratch brings further performance improvement. We leave a thorough investigation of this as future work.

**Additional Visualization** We provide additional text-to-image attention visualization results in Figure 2.

A man sits on a colorful man-drawn carriage, while another man stands beside it.

(a)

A woman and four children are crossing a busy street.

(b)

Figure 2: Additional visualization on text-to-image attention, comparing VILLA and UNITER.

## A.3 Downstream Tasks and Implementation Details

**Downstream Tasks** In VQA [4], GQA [5] and VCR [18], given an image and an input question, the model predicts an answer (or selects from a candidate pool). For NLVR$^2$ [13], given a pair of images and a natural language description, the model judges the correctness of the description based on the visual clues in the image pair. For Visual Entailment, we evaluate on SNLI-VE [15], where the model predicts whether a given image semantically entails a given sentence. For Referring Expression (RE) Comprehension, we evaluate on RefCOCO, RefCOCO+, and RefCOCOg datasets [17], where given a text description, the model selects the described region from a set of image region proposals. Models are evaluated on ground-truth objects and detected proposals. For Image-Text Retrieval (ITR), we consider both image retrieval and text retrieval on Flickr30k dataset.

For all the tasks except RE Comprehension, we extract the joint V+L embedding from the [CLS] token, and apply a multi-layer perceptron (MLP) for prediction. For RE Comprehension, we use MLP to compute the region-wise alignment scores. During the finetuning stage, ITR is formulated as a ranking problem, with triplet loss used for modeling training and hard negatives applied to boost performance [6]. All the other tasks can be formulated as a classification problem, using cross-entropy loss for model training. For VCR [18], second-stage pre-training with VCR training data was proven useful in [3]. Therefore, for VCR downstream experiments, we further apply 60k steps of second-stage adversarial pre-training.

**Probing Analysis** The visual coreference task aims to predict whether there is a link between an image region and a noun phrase in the sentence that describes the image. In addition, each coreference link in the dataset is annotated with a label. Through this task, we can find out whether the coreference knowledge can be captured by the attention trace. To achieve this goal, for each data sample in the Flickr30k Entity dataset, we extract the encoder's attention weights for all the 144 heads. Note that noun phrases typically consist of two or more tokens in the sequence. Thus, we extract the maximum attention weight between the image region and each word of the noun phrase for each head. The maximum weight is then used to evaluate which head identifies visual coreference.

Similarly, the visual relation task aims to identify and classify the relation between two image regions. The Visual Genome dataset is used for this task, which contains 1,531,448 relations. To reduce the imbalance in the number of relations per relation type, we randomly select at most 15,000 relation

| Task | Model | Batch Size | Grad. Accu. | Lr. | Training Steps | Warm-up Steps | Adv. Lr. | Adv. Weight |
|------|-------|-----------|-------------|-----|----------------|---------------|----------|-------------|
| VQA | VILLA$_{BASE}$ | 5120 | 5 | 8e-5 | 6000 | 600 | 1e-3 | 1.5 |
| | VILLA$_{LARGE}$ | 3072 | 8 | 5e-5 | 5000 | 500 | 1e-3 | 1.5 |
| VCR | VILLA$_{BASE}$ | 2000 | 10 | 6e-5 | 8000 | 800 | 1e-2 | 1.5 |
| | VILLA$_{LARGE}$ | 1000 | 20 | 6e-5 | 10000 | 1000 | 1e-1 | 1.0 |
| NLVR$^2$ | VILLA$_{BASE}$ | 2560 | 4 | 6e-5 | 3000 | 300 | 5e-4 | 1.5 |
| | VILLA$_{LARGE}$ | 1280 | 8 | 2e-5 | 5000 | 500 | 1e-2 | 1.5 |
| SNLI-VE | VILLA$_{BASE}$ | 4096 | 4 | 8e-5 | 5000 | 500 | 3e-3 | 2.0 |
| | VILLA$_{LARGE}$ | 4096 | 2 | 3e-5 | 4000 | 400 | 1e-3 | 1.5 |
| RefCOCO+ | VILLA$_{BASE}$ | 128 | 1 | 5e-5 | 8000 | 800 | 2e-3 | 1.0 |
| | VILLA$_{LARGE}$ | 96 | 1 | 4e-5 | 8000 | 800 | 1e-3 | 1.5 |
| RefCOCO | VILLA$_{BASE}$ | 128 | 1 | 4e-5 | 8000 | 800 | 5e-3 | 2.0 |
| | VILLA$_{LARGE}$ | 96 | 1 | 4e-5 | 10000 | 1000 | 1e-3 | 1.5 |
| RefCOCOg | VILLA$_{BASE}$ | 128 | 1 | 7e-5 | 12000 | 1200 | 2e-3 | 1.0 |
| | VILLA$_{LARGE}$ | 96 | 1 | 4e-5 | 8000 | 800 | 1e-3 | 1.0 |
| Flickr30k ITR | VILLA$_{BASE}$ | 32 | 32 | 5e-5 | 5000 | 500 | 1e-2 | 1.0 |
| | VILLA$_{LARGE}$ | 32 | 32 | 5e-5 | 5000 | 500 | 1e-2 | 1.0 |

Table 5: Hyper-parameter values used in our experiments.

pairs per type. Then, we perform similar probing analysis of the attention heads by examining the attention weights on ground-truth links.

**Implementation Details** Our models are implemented based on PyTorch. To speed up training, we use Nvidia Apex[1] for mixed precision training. All pre-training experiments are run on Nvidia V100 GPUs (16GB VRAM; PCIe connection). Finetuning experiments are implemented on the same hardware or Titan RTX GPUs (48GB VRAM). For large pre-training experiments, we use Horovod[2] and NCCL[3] for multi-node communication. All the hyper-parameter values used in experiments are listed in Table 5. And for all the experiments, we set the number of adversarial training steps to 3. We mostly follow the experimental settings in UNITER [3]. For more details on each downstream task finetuning, please refer to their Appendix. Since we mostly adopt their default hyper-parameters, and the only additional hyper-parameters we introduce are adversarial learning rate, number of adversarial steps, and the adversarial weight $\alpha$ in Eqn. 2 of the main text, the experimental results are fairly easy to reproduce.