[Reviews · NeurIPS 2020]

Review 1

Summary and Contributions: This paper introduces VILLA, a task-agnostic approach for training Transformer-based vision-language models with adversarial perturbations. The key idea is to add adversarial perturbations to the embedding space of visual and textual representations, and train the model to be invariant to these. An additional KL-term explicitly encourages the model's predictions with and without these adversarial perturbations to be similar. Experiments are conducted with UNITER (Chen et al.) as the base model, and adding VILLA on UNITER improves its performance on a host of tasks -- VQA, VCR, NLVR, SNLI-VE, RefCOCO, image-text retrieval -- achieving state-of-the-art results. The authors further conduct ablations to shed more light on how much VILLA helps 1) during pretraining vs. finetuning, 2) on the image vs. text domain. Finally, the authors also include some preliminary analysis to show that UNITER trained with VILLA learns better image-text correspondences (visual grounding for words) than the base UNITER model.

Strengths: This paper thoroughly explores a simple idea of adversarial training and demonstrates convincing results across a wide range of vision-language tasks. The proposed approach improves on the previous state-of-the-art by significant margins and thus makes an important empirical contribution.

Weaknesses: While this is a good paper overall demonstrating compelling results, it would be stronger and more complete if the following design choices are discussed / supported by empirical evidence: - How do adversarial perturbations in embedding space compare to those in pixel space? The latter follows more naturally from prior work in adversarial training. - What happens if adversarial perturbations are simultaneously added to both image and text domains, instead of one at a time? In addition to improving generalization performance, do these adversarial perturbations make the model more robust to adversarial attacks (where inputs, not embeddings, are adversarially perturbed)?

Correctness: To the best of my knowledge, the empirical evaluations sufficiently back claims made.

Clarity: The paper is clearly written and well-organized; it was a joy to read. Great work!This work appropriately situates itself in the context of prior work, and explores a complementary direction -- adversarial training for V&L models -- that hasn't been explored before in Transformer-based vision-language models.

Relation to Prior Work: This work appropriately situates itself in the context of prior work, and explores a complementary direction -- adversarial training for V&L models -- that hasn't been explored before in Transformer-based vision-language models.

Reproducibility: Yes

Additional Feedback:


Review 2

Summary and Contributions: Update: The rebuttal has not changed my (positive) opinion of the paper. The rebuttal, like the paper, seems strong. ------------- The paper performs large-scale adversarial training for vision + language representation learning. It demonstrates SOTA performance on 6 standard vision and language tasks. (My review is shorter than my average review because I think this work is solid, the paper is very well-written, and I don't have much else to say.)

Strengths: Clear, very well-written paper SOTA performance on 6 standard tasks Other systematic evaluation (ablation study to examine the effect of adversarial training only in the pre-training or only in the fine-tuning stage, evaluating adversarial pre-training across checkpoints, etc.) General idea, applied to a couple of different model architectures Submitted code, will release models > can we apply similar adversarial training techniques to V+L problems to improve model performance? Is a reasonable question that is worth asking and answering. Adding adversarial noise to the embeddings makes sense, and does make the approach more general (which is useful when dealing with multiple modalities).

Weaknesses: None that I can think of. More tasks, more base architectures, more ablations can always make the work stronger but I think this paper has more than sufficient empirical evaluation as is. One could say that this paper has limited novelty because it does not introduce a new model architecture or learning paradigm. But I think the question this paper asks and answers (listed above) is a useful one. The findings will be useful for the community. Authors have committed to releasing their models, which will also be useful for the community. So overall, I don't see any significant weaknesses.

Correctness: Everything seems correct

Clarity: Paper is very well-written and clear

Relation to Prior Work: Connections to prior work have been described well

Reproducibility: Yes

Additional Feedback: Out of curiosity: Did you experiment with y^ = f_theta(ximg + \deltaimg; xtxt + \deltatxt)? In general, if there were things the authors tried that didn't work out as well as expected, it would be worth describing those briefly in the paper. Readers might find that interesting + useful.


Review 3

Summary and Contributions: This paper presents a large-scale adversarial training for vision-and-language representation learning. Instead of adding adversarial perturbation into the input domain (image and text), this paper propose to add the noise into the embedding domain and extend FreeLB training scheme with two additional training objective: 1: label preserving attack and confidence preserving attack loss. The authors apply VILLA to current V+L models and achieve new state of the art on a wide range of tasks.

Strengths: It's very interesting to see the large-scale adversarial training for vision and language representation learning. The paper is well written and the result is very good. The presentation of the paper is also very nice.

Weaknesses: Besides the strength of the paper, I have some concerns about the paper. 1: The original goal of adversarial training is to avoid adversarial attacks. In this paper, the authors show that by adding adversarial perturbations into the embedding, the model can improve the performance on final downstream tasks. This is great, however, the paper didn't answer whether the proposed method can perform better in the adversarial attack? What is the connection between adding noise in embedding space and pixel/token space? There are multiple ways to test how the proposed method is more robust, for example: - Some downstream tasks focus on paraphrasing, there is a vqa-rephrasing dataset, and I am curious whether injecting the adversarial noise into the embedding space will lead to better performance on this dataset? (Cycle-Consistency for Robust Visual Question Answering). - What is the performance change when the model faces traditional adversarial attacks? (by adding perturbations into the pixels space and change tokens etc? ) 2: Since this model is trained based on the UNITER's saved checkpoints, it will have more optimization steps compared to UNITER. For a better comparison, the baseline UNITER model should have similar optimization steps. 3: The training seems a little bit tricky to me. Following the last question, in the supplementary materials, the author claims that most of the hyper-parameters are followed by UNITER, However, in table 5, the batch-size of the proposed model is very different from the one mentioned in UNITER paper. (VQA VILLA_BASE batch size 5120, while UNITER 10240, etc) I wonder how these hyperparameters selected and whether this is the same as in UNITER as the paper claimed?

Correctness: Yes

Clarity: Yes

Relation to Prior Work: yes

Reproducibility: Yes

Additional Feedback:


Review 4

Summary and Contributions: This paper presents a new large-scale adversarial training for vision-and-language (V+L) representation learning. The proposed framework consists of two training stages: task-agnostic adversarial pre-training and task-specific adversarial finetuning. Besides, it performs adversarial training in the embedding space of each modality and adopts the “free” adversarial training strategy, as well as KL-divergence-based regularization to guarantee large-scale cross-modal training. It conducts a wide range of V-L tasks and illustrates obvious improvements over existing methods.

Strengths: 1. This paper proposes a novel and brand new method of adversarial pre-training and adversarial finetuning for vision-and-language (V+L) representation learning. 2. This paper is well written and organized and it is fluent and clear for readers to understand. 3. This paper conducted comprehensive experiments on six different popular V-L tasks and show consistent improvements over existing methods.

Weaknesses: One slight concern is that visualizing only one example is not solid enough to embody the advantage of VILLA over UNITER.

Correctness: The paper is technically solid as it provides comprehensive theoretical and experimental supports for the proposed methods.

Clarity: The paper is perfectly written and organized.

Relation to Prior Work: This paper makes a comprehensive review of prior work and highlights the contributions it gains compared to the previous approaches.

Reproducibility: Yes

Additional Feedback: It’s a good job in every aspect.

[Author Response · NeurIPS 2020]

We thank all the reviewers for their insightful and encouraging comments. We confirm our promise to release all the code and pre-trained model checkpoints upon paper acceptance.

**To Reviewer 1, 3, and 4** **Q1**: *How do perturbations in embedding space compare to those in pixel/token space?*

**A1**: Most previous work studies adding perturbations in the pixel space for image classification task, and the general observation is that robustness is often at odds with generalization, *i.e.*, adversarial training hurts the performance on clean data. When dealing with V+L tasks, we observe that adding perturbations in the embedding space actually boosts the performance on clean data. We hope this observation can inspire future work on adding feature perturbations for adversarial training in different tasks.

Unlike pixels, tokens are discrete in nature. It still remains a challenging task to effectively craft adversarial examples in the token space without changing the semantic meaning of the original text. Since we only care about the *end results* of adversarial training on downstream tasks, adding embedding perturbations is a natural way to circumvent this obstacle.

Our embedding-based adversary is generic and flexible, as it can make arbitrary manipulations on embeddings, which is not possible if in the pixel/token space.

**Q2**: *What happens if adversarial perturbations are simultaneously added to both image and text domains?*

**A2**: Empirically, we observe that adding perturbations to both modalities simultaneously often reaches slightly worse performance than adding perturbations alternatively. This may be due to the fact that the injected noise is too much when perturbations are added simultaneously. Results are detailed in Table 1. More results on VILLA$_{\text{LARGE}}$ will also be included in the final version.

| Method | VQA | VCR (val) | | |
|---|---|---|---|---|
| | test-dev | Q→A | QA→R | Q→AR |
| VILLA$_{\text{BASE}}$ (simul.) | 73.15 | 75.50 | 78.60 | 59.56 |
| VILLA$_{\text{BASE}}$ (alter.) | **73.59** | **75.54** | **78.78** | **59.75** |

Table 1: Adding perturbations simultaneously vs. alternatively.

**Q3**: *Do the adversarial perturbations make the model more robust to adversarial attacks and paraphrases?*

**A3**: In order to test adversarial robustness, we need to perform adversarial attacks to existing V+L models. This V+L attack problem is largely unexplored in the literature. For example, how to reliably back-propagate the gradients from the multimodal Transformer to the CNN backbone to generate image adversaries is non-trivial. How to craft textual adversaries that align with the visual context is also challenging. In this paper, we mainly focus on improving model's generalization performance on clean data, leaving a more thorough investigation of adversarial attack and robustness as important future work. We are actively working on this.

| Method | VQA Acc. | |
|---|---|---|
| | Ori. | Rep. |
| BAN | 64.97 | 55.87 |
| BAN + CC | 65.87 | 56.59 |
| UNITER$_{\text{BASE}}$ | 70.35 | 64.56 |
| VILLA$_{\text{BASE}}$ | **71.27** | **65.35** |

Table 2: Results on VQA-Rephrasings.

To further address the reviewers' question, we have conducted additional experiments on the VQA-Rephrasings dataset to test the robustness of existing V+L models to paraphrases. For fair comparison, we have re-trained both UNITER and VILLA on the VQA training set only. Results are summarized in Table 2, where Ori. and Rep. denote the test set with original questions and their rephrasings. UNITER has already lifted up the performance by a large margin, and VILLA facilitates further performance boost.

**To Reviewer 4** **Q4**: *Unfair comparison with UNITER?*

**A4**: We would like to emphasize that the comparison with UNITER is a fair setting. As shown in L198-201 and Fig. 3(a), we carefully designed the experiments to make sure both UNITER and VILLA use the same number of optimization steps.

**Q5**: *The training seems a little bit tricky to me.*

**A5**: We confirm that we use exactly the same model configuration as UNITER. No hidden trick was used for training VILLA, and our results can be easily reproduced. The experimental settings in the original UNITER paper are not updated. For example, in their arXiv v1 version, UNITER-large achieved 73.82 on VQA test-std. This has been lifted to 74.02 with better optimized hyper-parameters. We use the authors' most recent hyper-parameter setting to perform experiments on VILLA. Also, in the official UNITER repo, the training batch size is indeed set to 5120 (`https://github.com/ChenRocks/UNITER/blob/master/config/train-vqa-base-4gpu.json`).

**To Reviewer 5** **Q6**: *One slight concern is that visualizing only one example is not solid enough.*

**A6**: We agree that visualizing only one example is not enough, and have provided Table 4 as a more systematic evaluation, in which the learned alignment has been carefully examined on the dataset level, rather than cherry-picking one single example. Also, we have provided another two examples in Appendix as well. We will clarify this.

[Meta-Review · NeurIPS 2020]

All four reviewers support acceptance for the contributions, notably the idea of using adversarial perturbations for training transformer-based vision-language models and successfully demonstrating this idea experimentally on a 6 standard vision&language tasks / datasets leading to SOTA results, all in a clearly written and organized paper. I agree to these observations and also recommend acceptance of this strong paper. The concerns the reviewers had, have been successfully addressed in the author response and I expect the authors will follow through with their promise to release all code and pre-trained models and revise the paper with correction, clarifications and additions from the rebuttal, including results on VILLA_LARGE and maybe add more additional qualitative examples in the appendix.